# Stabilizing the Exotic Carbonic Acid by Bisulfate Ion

**DOI:** 10.3390/molecules27010008

**Published:** 2021-12-21

**Authors:** Huili Lu, Shi-Wei Liu, Mengyang Li, Baocai Xu, Li Zhao, Tao Yang, Gao-Lei Hou

**Affiliations:** 1Master Kong Beverage R&D Center, Shanghai 201103, China; luhuili@masterkong.com.cn; 2China National Research Institute of Food & Fermentation Industries Co., Ltd., Beijing 100015, China; ff_nns@163.com; 3School of Light Industry, Beijing Technology and Business University, Beijing 100048, China; zhaol@btbu.edu.cn; 4MOE Key Laboratory for Non-Equilibrium Synthesis and Modulation of Condensed Matter, School of Physics, Xi’an Jiaotong University, Xi’an 710049, China; lmy0916@stu.xjtu.edu.cn

**Keywords:** carbonic acid, bisulfate ion, density functional theory calculations, molecular dynamics simulations

## Abstract

Carbonic acid is an important species in a variety of fields and has long been regarded to be non-existing in isolated state, as it is thermodynamically favorable to decompose into water and carbon dioxide. In this work, we systematically studied a novel ionic complex [H_2_CO_3_·HSO_4_]^−^ using density functional theory calculations, molecular dynamics simulations, and topological analysis to investigate if the exotic H_2_CO_3_ molecule could be stabilized by bisulfate ion, which is a ubiquitous ion in various environments. We found that bisulfate ion could efficiently stabilize all the three conformers of H_2_CO_3_ and reduce the energy differences of isomers with H_2_CO_3_ in three different conformations compared to the isolated H_2_CO_3_ molecule. Calculated isomerization pathways and ab initio molecular dynamics simulations suggest that all the optimized isomers of the complex have good thermal stability and could exist at finite temperatures. We also explored the hydrogen bonding properties in this interesting complex and simulated their harmonic infrared spectra to aid future infrared spectroscopic experiments. This work could be potentially important to understand the fate of carbonic acid in certain complex environments, such as in environments where both sulfuric acid (or rather bisulfate ion) and carbonic acid (or rather carbonic dioxide and water) exist.

## 1. Introduction

Carbonic acid, H_2_CO_3_, is a diprotic oxyacid and has long been considered as non-existing in isolated state [1]. Its importance in a range of fields, including astrophysics, astrobiology, astrochemistry, geography, and biochemistry, has been well-recognized [2,3,4,5,6,7]. It is of great significance in regulating blood pH, in adjusting ocean acidification, and in the dissolution of carbonate minerals [8]. In recent years, its potential as an interstellar molecule on the Martian surface, comets, and icy grain mantles has also attracted a lot of attention since both decomposition components of carbonic acid (i.e., carbon dioxide and water) have been discovered in those environments [4,5,9].

The non-existence of carbonic acid in isolated state was thought to be due to the fact that it is thermodynamically favorable to decompose into water and carbon dioxide, although theoretical calculations indicated that it might be kinetically trapped by a potential well [10,11]. The decomposition of carbonic acid could be efficiently promoted by a variety of species, such as water, ammonia, sulfuric acid, organics, and cloud particle surfaces. That could be one reason why isolated H_2_CO_3_ has escaped direct detection for a long time [11,12,13,14,15,16,17]. In 1987, Schwarz and co-workers reported the first observation of isolated H_2_CO_3_ via neutralization-reionization mass spectrometry of heated ammonium bicarbonate (NH_4_HCO_3_) vapor [1]. More than 20 years later, Endo and co-workers first spectroscopically detected both the *cis-trans* H_2_CO_3_ molecule in 2009 and the *cis-cis* H_2_CO_3_ molecule in 2011 using Fourier-transform microwave spectroscopy; the two H_2_CO_3_ isomers were produced via supersonic expansion and pulsed electric discharge of a carbon dioxide and water mixture [18,19]. In 2011, Bernard et al. also detected the two isomers of H_2_CO_3_ with the *cis-trans* isomer being less stable than the *cis-cis* one by about 4 kJ/mol and showed that H_2_CO_3_ is stable at temperatures above 200 K [20]. Later in 2014, Schreiner and co-workers reported a novel and general approach to prepare H_2_CO_3_ and its monomethyl ester (CH_3_OCO_2_H) through gas-phase pyrolysis of di-tert-butyl and tert-butyl methyl carbonate, respectively [21]. Furthermore, they verified that the previously thought polymorph of solid carbonic acid *α*-H_2_CO_3_ actually belongs to the carbonic acid monomethyl ester, and the β-polymorph is true H_2_CO_3_. Very recently, Ioppolo et al. provided evidence that β-H_2_CO_3_ should be ubiquitously present in space on the surface of CO_2_- and H_2_O-rich ices. They also observed unique spectral features of *γ*-H_2_CO_3_, deserving future search in the coldest regions of the interstellar medium [22]. Moreover, Wang et al. found that carbonic acid can even be formed from CO_2_ on ice in the absence of high-energy irradiation and without protonation by strong acids, implying its potential role in the upper troposphere in cirrus clouds [23].

Hydrogen bonding is an effective way to stabilize otherwise unstable and uncommon structures [24,25,26,27,28,29,30,31,32,33,34]. For example, Hou et al. have performed a series of studies on bisulfate ion-containing complexes, showing that hydrogen bonding interactions can alter the protonation pattern in the complex, which violates the gas-phase proton affinity prediction [24,27,35,36,37,38]. This motivated us to wonder whether an appropriate partner such as bisulfate ion could also stabilize the exotic conformers of H_2_CO_3_ molecule. In fact, Thomas et al. has recently measured the infrared spectrum of the carbonic acid–fluoride complex anion in helium nanodroplets and found remarkable stability of F^−^(H_2_CO_3_) with carbonic acid in a *trans-trans* conformation [39]. Similar work on the whole series of carbonic acid–halide complexes, X^–^(H_2_CO_3_) (X = F, Cl, Br, and I), has been followed by Zhang et al. through joint photoelectron spectroscopy and ab initio calculations [40]. Here in this work, we theoretically studied the carbonic acid–bisulfate ion complex by using density functional theory (DFT) calculations and molecular dynamics simulations. The stability and bonding property of the complex have been explored.

## 2. Theoretical Methods

Geometry optimizations of all minima and transition states on the potential energy surface of the anionic molecular complex [H_2_CO_3_·HSO_4_]^−^ were performed by using DFT calculations. Specifically, M06-2X functional and aug-cc-pVTZ basis set were employed, as previous studies showed that this combination could provide reliable results for hydrogen bonded, bisulfate ion-containing complexes [24,27,35,36]. Harmonic frequency analysis was conducted at the same level of theory. All the quantum chemical calculations were performed with Gaussian09 program [41]. The natural bond orbital (NBO) analyses were carried out at M06-2X/aug-cc-pVTZ level using NBO 3.1 as implemented in Gaussian09. Ab initio molecular dynamics (AIMD) simulations using the Nosé-Hoover heat bath scheme with the average temperature of the system at 300, 400, 500, and 1000 K were performed for the most stable isomer, and an average temperature of 300 K for the second most stable isomer, with the Vienna ab initio simulation package (VASP) [42,43,44]. The PBE functional was used for the exchange-correlation functional [45]. A unit cell of 30×30×30 Å has been employed to avoid spurious interaction in space and the reciprocal space was represented by the Gamma point. The plane wave cut-off energy in the wave vector K space was 520 eV with the convergence criteria for the energy as 1 × 10^−4^ eV. Quantum theory of atoms in molecules (QTAIM) [46] analyses were also performed with Mutiwfn software [47] to gain insights into the H-bonding interactions in [H_2_CO_3_·HSO_4_]^−^.

## 3. Results and Discussion

Figure 1 presents the optimized low-lying isomers of [H_2_CO_3_·HSO_4_]^−^. To better understand the changes of each component in [H_2_CO_3_·HSO_4_]^−^ upon complexation, we provide the optimized three conformers of carbonic acid, bisulfate ion, and sulfuric acid in Appendix A. It can be seen that the optimized structures of the three conformers are in good agreement with previous studies, and their relative stabilities are also consistent with previous calculations at a higher level of theory CCSD(T)/cc-pVQZ [10,11,18], verifying the reliability of the theoretical method utilized in the current work. For [H_2_CO_3_·HSO_4_]^−^, we have obtained six stable isomers, among which isomers **2a** and **2b** could be considered as enantiomers, and isomers **3a** and **3b** could also be considered as enantiomers. While isomers **1**, **3**, and **4** may be regarded as HSO4^–^·H_2_CO_3_, isomer **2** is better recognized as H_2_SO4·HCO_3_^−^, which is only about 2 kJ/mol higher in energy than isomer **1**. Isomer **2** is unexpected solely from gas-phase proton affinity prediction and its stability can be attributed to the formation of two strong hydrogen bonds and highly delocalized extra electrons according to previous findings by Hou et al. [24,27,36,37,38]. Such electron delocalization can be partly reflected by the highest occupied molecular orbitals (HOMOs) as presented in Appendix A. Since we are mainly interested in understanding if bisulfate ion could stabilize carbonic acid molecule, in the following we will focus on isomers **1**, **3**, and **4**.

It is interesting to note that in isomers **1**, **3a**, **3b**, and **4**, carbonic acid moiety is in the form of *trans-trans*, *cis-trans*, *cis-cis*, and *cis-trans*, respectively. For bare carbonic acid, *cis-cis* conformer is the most stable one, and *cis-trans* and *trans-trans* are higher in energy by 6.1 and 41.2 kJ/mol, respectively. Upon complexation with bisulfate ion, the isomer with carbonic acid moiety in *trans-trans* becomes to be the most stable one, which could be due to the formation of two equivalent hydrogen bonds. The energy differences of isomers with carbonic acid moiety in *cis-trans* and *cis-cis* relative to the most stable *trans-trans* one are much smaller compared in the bare case, suggesting the efficient stabilization of carbonic acid molecule by bisulfate ion. The smaller energy difference implies that these isomers may convert to each other upon gaining sufficient perturbation (for example, by thermionic heating or photoabsorption).

In Figure 2, we present the isomerization pathways between the different isomers of [H_2_CO_3_·HSO_4_]^−^ and the energy barriers needed to be overcome. The calculations show that the isomerization between isomers **2a** and **2b**, isomers **3a** and **3b** involves the rotation of −OH group in bicarbonate or carbonic acid, and the barriers are almost equal to be 45.4 and 46.4 kJ/mol, respectively. Such relatively high energy barriers may preclude the isomerization between the enantiomers of **2** and **3**, and they may all exist at finite temperatures. Isomerization between isomers **2a** and **3a**, isomers **2b** and **3b** has low barriers of only about 8 kJ/mol, indicating that the proton translocation between the two negatively charged oxygen atoms should not be difficult, as found in HCOO^−^·H^+^·HSO_4_^–^ [36,48]. Visually, it seems isomerization between isomer **4** and isomer **3b** is more likely due to the fact that they have same hydrogen bond pattern between the two partners. However, due to the low energy barrier of proton translocation, our calculations show that isomerization happens between isomers **4** and **2b** via a transition state **ts42b** in which a proton translocates to bisulfate ion moiety simultaneously with the H transfer. This isomerization process has a large barrier of about 140 kJ/mol which is likely mainly resulted from the H transfer, making the isomerization between isomers **4** and **2b** not likely under ambient conditions. Conversion between isomers **1** and **4** is a stepwise process via an intermediate state **in14**. The first step is the H transfer from the −SOH moiety to the O of HSO_4_^–^ that connects to the H_2_CO_3_ via a OˑˑˑH–O hydrogen bond; this step has a high energy barrier of ~125 kJ/mol. The second step is the swing of the H of one −OH of the H_2_CO_3_ with a comparably lower barrier of about 56 kJ/mol. This stepwise process has been confirmed by intrinsic reaction coordinate (IRC) scan (Appendix A). The large barrier of the first step renders the isomerization to be only possible at elevated temperatures or heating conditions.

To further evaluate the thermodynamical stability of the different isomers of [H_2_CO_3_·HSO_4_]^−^, we performed AIMD simulations for isomers **1** and **2a**, as presented in Figure 3. It can be seen that after 10,000 simulation steps (10 picoseconds), the structures of both isomers only show slight deformation compared to the ground state at 0 K, suggesting their high thermodynamic stability. Temperature and energy fluctuate over time in a steady-state, and the difference of the maximum and the minimum energies (ΔE) of isomer **1** and isomer **2a** are only about 0.5 eV, confirming their high-temperature stability [49,50]. We further simulated isomer **1** at higher temperatures of 400, 500, and 1000 K, all presenting its good stability (see animations in Appendix A).

As stated above, the high stability of those isomers comes from hydrogen bond formation and extra electron delocalization. The electron delocalization could partly be reflected by the HOMOs as shown in Appendix A and partly by the charge distributions as summarized in Appendix A. QTAIM is a useful approach to evaluate the nature of hydrogen bond and its strength in addition to the geometrical analysis [37,46]. According to Bader’s theory, the critical points (3,−3) are the nuclear positions where the charge density is local maximum in all directions, and the BCPs (3,−1) correspond to the second-order saddle points, for which two eigenvalues of the charge density Hessian matrix are negative and one is positive. The BCPs and the nuclei are connected by the maximum charge density paths. Appendix A presents the molecular graphs showing all the BCPs. The electron density (*ρ*) and its Laplacian (∇^2^*ρ*) at BCP are closely related to bonding strength and bonding type, respectively; the potential energy density (V(r)), gradient kinetic energy density (G(r)), and electronic energy density (K(r)) are highly correlated with the hydrogen bond energies. These values are summarized in Table 1.

From Table 1, it can be seen that the electron density distribution (*ρ*) and its Laplacian (∇^2^*ρ*) at BCPs of the two hydrogen bonds of isomers **1** and **2** are almost identical, while those of isomers **3** and **4** differ a lot, consistent with the structural analyses. Such difference can also be seen from the values of electronic energy density K(r), whose magnitude is related to the hydrogen bond strength. In 2019, Emamian et al. investigated a series of hydrogen bonded complexes and fitted an equation E_HB_ = −332.34 × *ρ*_BCP_ − 1.0661 to estimate the hydrogen bond energy (E_HB_) for hydrogen bonded ionic complexes. This equation gives a mean absolute percentage error (MAPE, in kcal/mol) of 10.0%, and *ρ*_BCP_ is the electron density at the BCP of hydrogen bonds in a.u., and E_HB_ is in kcal/mol [52]. The hydrogen bond energies estimated in this way are summarized in Table 1, and it is shown clearly that the calculated hydrogen bond strengths are consistent with the optimized geometric structures (i.e., the shorter O–HˑˑˑO distance, the higher hydrogen bond energy).

Temperature-controlled infrared spectroscopy is a useful experimental approach to probe the proton translocation dynamics as demonstrated by Johnson and co-workers [53,54] and very recently also by von Helden and co-workers [39,48]. To guide future infrared spectroscopic experiments, we simulated the harmonic vibrational spectra of the six isomers of [H_2_CO_3_·HSO_4_]^−^, as presented in Figure 4.

In isomer 1, there is a very weak vibrational mode at ~3680 cm^−1^ corresponding to the O–H stretching of the HSO_4_^–^ moiety, and the strong vibration at ~3180 cm^−1^ is the symmetric double O–H stretching of the H_2_CO_3_ moiety. The spectra of isomers **2a** and **2b** are almost identical, as they are enantiomers; the vibrations at ~2230 and 2410 cm^−1^ are the asymmetric and symmetric stretching of the two shared H^+^ between SO_4_^2–^ and HCO_3_^–^ moieties. Similarly, the spectra of isomers **3a** and **3b** are almost identical; the weak mode at ~3700 cm^−1^ is the free O–H stretching of the H_2_CO_3_ moiety, and the two strong modes at ~1810 and 3270 cm^−1^ are the shared H^+^ translocation and hydrogen bonded O–H stretching of the HSO_4_^–^ moiety. In the simulated spectrum of isomer **4**, the vibrations at ~2130, 3590, and 3700 cm^−1^ are the shared H^+^ translocation, hydrogen bonded O–H stretching of the HSO_4_^–^ moiety, and the free O–H stretching of the H_2_CO_3_ moiety. Those vibrational vectors are included in Appendix A. It should be mentioned that more accurate description of the vibrations in those structures requires more advanced theoretical treatments including both anharmonic effects and nuclear quantum effects, and is beyond the scope of current study [48].

## 4. Conclusions

In summary, we theoretically explored the potential energy surface of the anionic [H_2_CO_3_·HSO_4_]^−^ complex to investigate if bisulfate ion is capable to stabilize the exotic conformers of carbonic acid. Our results showed that compared to the bare H_2_CO_3_ molecule, the energy differences between the different isomers of [H_2_CO_3_·HSO_4_]^−^ complex with H_2_CO_3_ moiety in the conformation of *trans-trans*, *cis-trans*, and *cis-cis* become smaller, suggesting that bisulfate ion can efficiently stabilize the three conformers of H_2_CO_3_. Notably, for bare H_2_CO_3_, the *cis-cis* conformer is the most stable, while in [H_2_CO_3_·HSO_4_]^−^ the isomer with H_2_CO_3_ in *trans-trans* becomes the most stable due to the formation of two strong hydrogen bonds. Combining quantum theory of atoms in molecules topological analysis and a fitted empirical equation, we estimated the strengths of the hydrogen bonds. The calculated isomerization pathways between the different isomers of [H_2_CO_3_·HSO_4_]^−^ and also ab initio molecular dynamics simulations of the first two low-lying isomers revealed that these isomers could coexist at finite temperatures. We further simulated their harmonic infrared spectra to facilitate future infrared spectroscopic experiments to fully understand the proton translocation dynamics in this interesting complex.

## Figures and Tables

**Figure 1 molecules-27-00008-f001:**
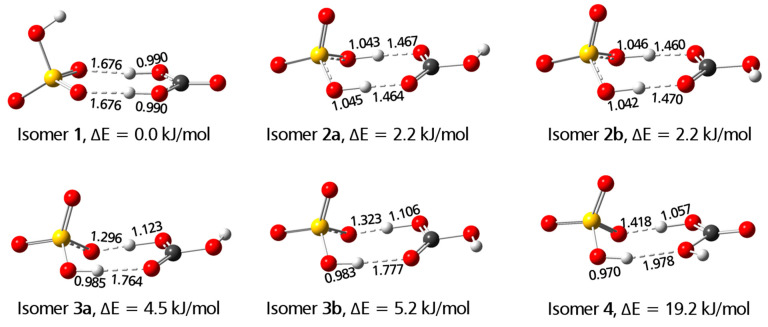
Optimized low-lying isomers of [H_2_CO_3_·HSO_4_]^−^ at M06-2X/aug-cc-pVTZ level of theory. Isomers **2a** and **2b** are energy degenerate, and isomers **3a** and **3b** are also energy degenerate.

**Figure 2 molecules-27-00008-f002:**
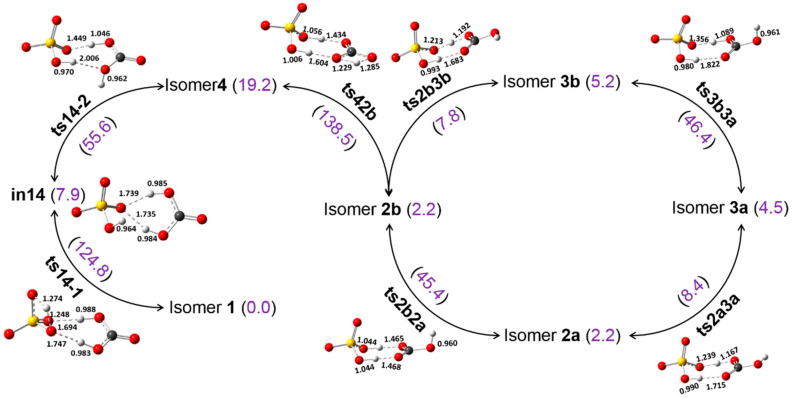
Calculated isomerization pathways between the different isomers of [H_2_CO_3_·HSO_4_]^−^. The optimized structures of transition states are provided, and the energy barriers are in kJ/mol in parenthesis.

**Figure 3 molecules-27-00008-f003:**
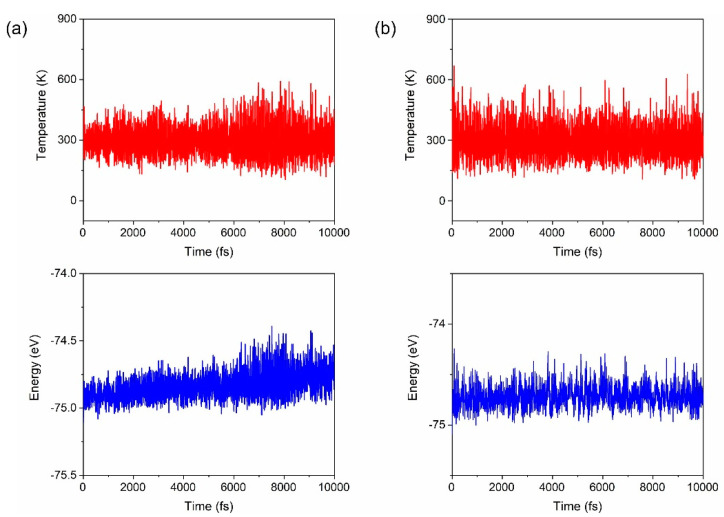
Total energy and temperature fluctuations with respect to time of the AIMD simulation for isomers **1** (**a**) and **2a** (**b**) both at 300 K.

**Figure 4 molecules-27-00008-f004:**
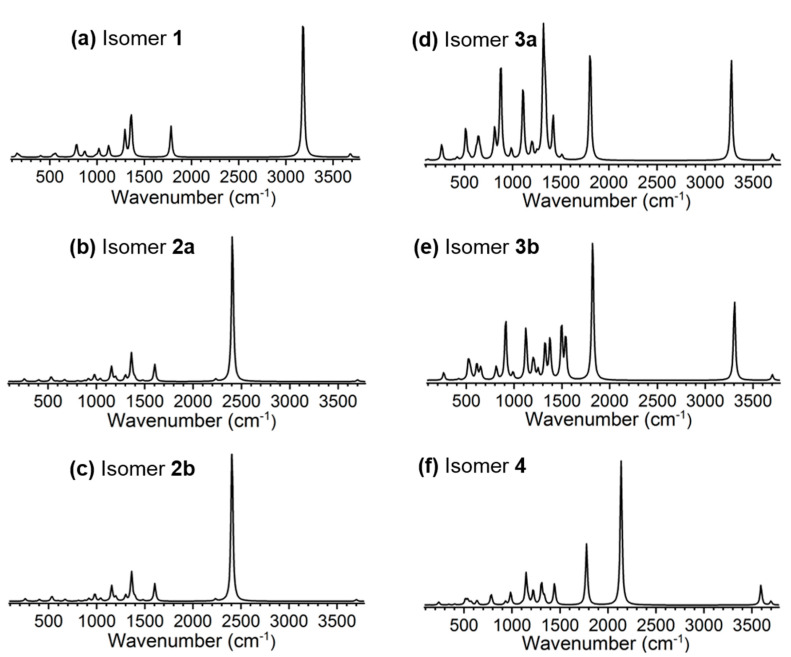
Simulated harmonic infrared spectra of the six isomers (**a**–**f**) of [H_2_CO_3_·HSO_4_]^−^ at M06-2X/aug-cc-pVTZ level of theory. The calculated vibrational frequencies have been broadened using Gaussian line shape of 12 cm^−1^ full width half maximum, and a scaling factor of 0.956 [55] has been applied.

**Table 1 molecules-27-00008-t001:** Electron density (*ρ*), Laplacian (∇^2^*ρ*), potential energy density (V(r)), gradient kinetic energy density (G(r)), and electronic energy density (K(r)) at BCPs of the hydrogen bonds in the different isomeric structures of [H_2_CO_3_·HSO_4_]^−^. All units are in a.u, except that E_HB_ is in kcal/mol.

[H_2_CO_3_·HSO_4_]^−^	*ρ* (10^−2^) ^a^		∇^2^*ρ* (10^−2^)		V(r) (10^−2^)		G(r) (10^−2^)		K(r) (10^−2^)		E_HB_	
	I ^b^	II	I	II	I	II	I	II	I	II	I	II
Isomer **1**	4.696	4.695	10.923	10.923	−4.960	−4.959	3.845	3.845	1.115	1.114	−16.67	−16.67
Isomer **2a**	8.397	8.454	7.844	7.692	−10.000	−10.087	5.980	6.005	4.019	4.082	−28.97	−29.16
Isomer **2b**	8.448	8.403	7.699	7.837	−10.080	−10.009	6.002	5.984	4.077	4.025	−29.14	−28.99
Isomer **3a**	12.889	3.760	−6.519	10.678	−17.922	−3.696	8.146	3.183	9.776	0.513	−43.90	−13.56
Isomer **3b**	11.976	3.629	−1.754	10.665	−16.088	−3.535	7.825	3.101	8.263	0.435	−40.87	−13.13
Isomer **4**	9.311	2.160	6.644	8.408	−11.505	−1.786	6.583	1.944	4.922	−0.158	−32.01	−8.24

^a^ Positive values of *ρ* indicate closed-shell interactions between two hydrogen bonded atoms according to Koch and Popelier [51]. ^b^ See Appendix A for the notation of hydrogen bonds I and II in each isomer.

## Data Availability

Not available.

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
