# Peer review of "Stabilizing the Exotic Carbonic Acid by Bisulfate Ion"

_molecules, 2021, doi:10.3390/molecules27010008_

Round 1

Reviewer 1 Report

In this paper, a theoretical study of the ionic complex (H2CO3·HSO4) – is carried out at the M06-2X level using a DZ Dunning basis set, molecular dynamics and QTAIM analysis. Five isomer structures are localized and their isomerization pathways, barriers and IR spectra are studied.

The authors have used M06-2X/aug-cc-pVTZ level of calculation. In Theoretical Methods section, they justify the use of this functional because it gives reliable results (refs. 24, 27 and 35) for other systems containing hydrogen bonds. However, we can find in the literature  authors that justify the usage of other functionals. In Boese’s paper (ChemPhysChem 2015, 16, 978 – 985), for instance, the author concludes that at least a triple-z quality that includes diffuse functions should be used as basis set. Regarding functionals, Boese concludes that Truhlar functionals give small errors when long-range dispersion is included (M06+D3 functional or similar) and accurate results are obtained using MP2.

My main objection goes further. DFT methods are single-determinant based and it is difficult to justify the pathway from the isomer 1 to isomer 4 where it si necessary to break an OH bond (close to S) to form a new bond OH (close to C). For a realistic description it is necessary to use multiconfigurational methods.

In spite of the above comments, I think that the paper is well written and the conclusions can be useful.

Reviewer 2 Report

Please see attached PDF.

Round 2

Reviewer 1 Report

I consider that the authors have satisfactorily answered my comments.